# HSP70/DNAJ Family of Genes in the Brown Planthopper, *Nilaparvata lugens*: Diversity and Function

**DOI:** 10.3390/genes12030394

**Published:** 2021-03-10

**Authors:** Xuan Chen, Ze-Dong Li, Dan-Ting Li, Ming-Xing Jiang, Chuan-Xi Zhang

**Affiliations:** 1Institute of Insect Science, Zhejiang University, Hangzhou 310058, China; xuanchen@zju.edu.cn (X.C.); lizedong@zju.edu.cn (Z.-D.L.); 11516079@zju.edu.cn (D.-T.L.); 2State Key Laboratory for Managing Biotic and Chemical Threats to the Quality and Safety of Agro-Products, Key Laboratory of Biotechnology in Plant Protection of MOA of China and Zhejiang Province, Institute of Plant Virology, Ningbo University, Ningbo 315211, China

**Keywords:** *Nilaparvata lugens*, HSP70, DNAJ

## Abstract

Heat shock 70kDa proteins (HSP70s) and their cochaperones DNAJs are ubiquitous molecular chaperones, which function as the “HSP70/DNAJ machinery” in a myriad of biological processes. At present, a number of HSP70s have been classified in many species, but studies on DNAJs, especially in insects, are lacking. Here, we first systematically identified and characterized the HSP70 and DNAJ family members in the brown planthopper (BPH), *Nilaparvata lugens*, a destructive rice pest in Asia. A total of nine HSP70 and 31 DNAJ genes were identified in the BPH genome. Sequence and phylogenetic analyses revealed the high diversity of the NlDNAJ family. Additionally, spatio-temporal expression analysis showed that most NlHSP70 and NlDNAJ genes were highly expressed in the adult stage and gonads. Furthermore, RNA interference (RNAi) revealed that seven NlHSP70s and 10 NlDNAJs play indispensable roles in the nymphal development, oogenesis, and female fertility of *N. lugens* under physiological growth conditions; in addition, one HSP70 (*NlHSP68*) was found to be important in the thermal tolerance of eggs. Together, our results in this study shed more light on the biological roles of HSP70/DNAJ in regulating life cycle, coping with environmental stresses, and mediating the interactions within, or between, the two gene families in insects.

## 1. Introduction

Facing a complicated and changing environment, insects have evolved the ability to maintain cellular and protein homeostasis, which enhances their development and survival during normal growth and upon exposure to stress. In insects, one of the strategies to adapt to the changing environments is to regulate the synthesis of heat shock proteins (HSPs), also known as the molecular chaperones [1,2]. HSPs are ubiquitously expressed in most cell types and almost all organisms [3]. HSPs can be divided into several classes according to their molecular mass, structural characteristics, and functions, including HSP100, HSP90, HSP70, HSP60, HSP40, and small HSPs [1]. Among those HSP families, heat shock 70 kDa proteins (HSP70s) are the most ubiquitous and evolutionarily conserved members [3,4,5,6]. HSP70 is not only essential for protection against stressful conditions, such as high/cold temperature, starvation, and anoxia, but is also important for fundamental housekeeping functions in non-stressed cells. To be specific, the multiple roles of HSP70 include protein folding, degradation, transportation, protein–protein interactions (PPIs), and prevention and dissolution of protein aggregates [1,7,8,9]. Genes in the HSP70 family can be further classified into two subfamilies, namely, HSP70 and HSP110 (also referred to as HSPH) [4], which function in the cytoplasm, mitochondrion, and endoplasmic reticulum (ER) [10]. In addition, cytoplasmic HSP70s can be further categorized into constitutively expressed proteins (HSC70s) and stress-inducible proteins (HSP70s).

However, HSP70s never function alone; instead, they function as the “HSP70 machinery” [8]. HSP70s, along with their well-characterized cochaperones HSP40s (also known as J proteins or DNAJ), constitute a versatile chaperone machinery for fulfilling functional diversity [11]. Compared with HSP70s, HSP40s are less conserved and show great diversity, in terms of their sequences and structures. Apart from the signature J domain that modifies the ATPase activity of HSP70s, J proteins often have distinctive additional domains and motifs. Consequently, J proteins are further categorized into three subclasses (class I–III, also known as class A, B, and C, respectively) [12]. The DNAJA class includes an N-terminal J domain, a Gly and Phe-rich region, four repeats of the CxxCxGxG type zinc finger motif, and a C-terminal peptide-binding domain. In addition, proteins classified as DNAJB are similar to those in the DNAJA class, but they lack the zinc-finger domain. The DNAJC class is the most diverse, and comprises a J domain and various other functional modules [13,14]. Generally, the number of HSP70s is rather limited compared with the DNAJ proteins. For example, in humans, there are only 11 HSP70s, but at least 41 J proteins. In *Drosophila melanogaster* and *Saccharomyces cerevisiae*, 14 Hsp70s and 22 J proteins, respectively, have been identified [8,15,16]. In addition, it has been reported that there are 18 Hsp70s and 116 J proteins in *Arabidopsis thaliana* [17]. As a result, it is suggested that multiple J proteins often function with a single HSP70 [18].

Previous studies have identified a number of HSP70s in different species; however, most studies in insects have been restricted to the HSP structural characteristics and expression pattern analysis, whereas the physiological or biological functions of insect HSPs remain largely unknown. Furthermore, there are few reports on the DNAJ gene family in insects, especially in terms of systemic analysis.

*Nilaparvata lugens* is one of the most destructive rice pests in Asian countries. *N. lugens* is a typical migrant insect, whose growth, development, distribution, and population abundance are greatly affected by temperature, and this pest is often used to forecast population outbreaks [19,20]. Previous studies have discovered that a large number of HSP genes are the most co-regulated at high temperature (37 °C), indicating that HSPs are important in the thermal tolerance of this species [21]. Furthermore, RNA interference (RNAi) is a powerful approach for inducing the loss-of-function in brown planthopper (BPH), making this organism an attractive model for investigating the in vivo correlation of gene expression patterns with HSPs functions. Recently, Lu and his colleagues cloned one HSP70 and one HSC70 gene in *N. lugens,* and analyzed their responses to temperature and insecticide stresses. As a result, HSP70 was not affected by heat shock, but was remarkably induced by imidacloprid; in addition, HSC70 showed differential expression in response to thermal stress in two wing morphs [22,23]. However, information about other HSP70s and the large number of their cochaperone DNAJs, including gene family constitution, biological functions, and protein interactions (PIs) between HSP70s and DNAJs, is lacking.

In the present study, we identified and characterized the HSP70 and DNAJ family members of BPH. The gene architectures, phylogenetic relationships, and expression patterns under both normal and temperature stressed conditions were analyzed. In addition, these genes were comprehensively screened by RNAi to uncover new biological roles of these molecular chaperones in growth, development, and thermal tolerance, which shed more light on the HSP70/DNAJ machinery.

## 2. Materials and Methods

### 2.1. Insects, Growth Conditions, and Stress Treatments

BPHs used in this study were originally obtained from Hangzhou, China. The insects were fed on fresh rice seedlings (strain: Xiushui 134) in a walk-in chamber at 27 ± 0.5 °C, 60 ± 5% relative humidity (RH) and in 16 h/8 h light/dark cycle conditions.

To investigate the heat shock responses of target HSP genes, BPHs were collected and exposed to heat and cold stresses. In brief, BPHs grown under normal conditions were transferred to an artificial climate incubator maintained at 37 °C or 4 °C for 1 h. Then, the treated insects were used for subsequent experiment.

### 2.2. Whole-Genome Identification of the NlDNAJ and NlHSP70 Family Members in BPH

All the candidate genes of BPH were retrieved from the genome and transcriptome database of BPH. The amino acid sequences of the DmHSPs were selected as the query sequences to identify NlHSPs. In addition, the Pfam (http://pfam.xfam.org/ version 33.1 (accessed on 5 September 2020)) and SMART (http://smart.embl-heidelberg.de/ (accessed on 5 September 2020)) search tools were used in combination to annotate the putative domains.

### 2.3. Sequence Characterization and Phylogenetic Tree Construction

The open reading frames (ORFs) were predicted using the Softberry website. The online program Cell-PLoc-2 (http://www.csbio.sjtu.edu.cn/bioinf/Cell-PLoc-2/ (accessed on 7 September 2020)) was employed to predict the subcellular localization of proteins. In addition, the isoelectric points (PIs) and molecular weights (MWs) were estimated using the Compute pI/Mw tool from ExPASy (http://web.expasy.org/compute_pi (accessed on 10 September 2020)). Exon-intron organization was analyzed by aligning the genomic DNA with the full-length cDNA of HSP70 and DNAJ genes. Moreover, the Gene Structure Display Server (GSDS 2.0 http://gsds.gao-lab.org/ (accessed on 11 October 2020)) was utilized to analyze the intron and exon construction of each gene. Amino acid sequence alignments were carried out with ClustalX v1.83 and GeneDoc v2.7.0. Phylogenetic trees were constructed by the MEGA 7 program according to the Neighbor-joining method, and with bootstraps of 1000 replicates. The HSP70 and DNAJ homologs derived from other species are listed in Appendix A.

### 2.4. RNA Isolation and Expression Pattern Analysis of NlDNAJ and NlHSP70 Genes

First of all, BPHs were collected at different developmental stages (eggs, second instar nymphs, fifth instar nymphs, and adults at 24 h after emergence), then BPHs at each stage were further divided into three groups and reared at 4 °C, 27 °C, and 37 °C for 1 h, respectively. Thereafter, the total RNA was extracted using RNAiso Plus (Takara, Kyoto, Japan). Similarly, the total tissue RNA from integuments, fat bodies, guts, wing buds, testes, and ovaries was extracted to analyze the expression levels.

Then, 1 μg RNA was prepared from each sample to synthesize the first-strand cDNA using the HiScript^®^ II Q RT SuperMix (Vazyme, Nanjing, China). Thereafter, real-time quantitative PCR (RT-qPCR) was carried out using a SYBR Color qPCR Master Mix kit (Vazyme, Nanjing, China) to quantify target gene expression levels. The reaction was conducted on a CFX96TM real-time PCR detection system (Bio-Rad, Hercules, CA, USA) under the following conditions: denaturation for 3 min at 95 °C, followed by 40 cycles at 95 °C for 10 s, and 60 °C for 30 s. Specific primers were designed using Primer Premier 6.0 software (Appendix A), with the housekeeping gene 18S rRNA being an internal control. Then, the relative expression levels of target genes were normalized to that of the reference gene by the 2^−ΔΔCt^ method. Results were presented in the form of a heatmap created using the MultiExperiment Viewer (MeV) software v 4.8.1.

### 2.5. RNAi Experiment under Normal Growth Condition

The purified DNA sequences were utilized as templates to synthesize the double-stranded RNAs (dsRNAs) using a T7 high-yield transcription kit (Vazyme, Nanjing, China) in accordance with the manufacturer’s instructions. The specific primers containing the T7 RNA polymerase promoter at both ends were designed, as listed in Appendix A, with *GFP* gene being a negative control.

The RNAi experiment was performed according to the previously reported method [24,25]. In brief, a certain amount of dsRNA from each target gene was microscopically injected into the mesothorax of the 3rd instar nymphs (approximately 25 ng) or the newly emerged females (approximately 100 ng). At 3 days after injection, 10 and 5 BPHs from two different developmental groups were collected to determine the RNAi efficiency by RT-qPCR, respectively. The remaining insects were maintained for phenotype observation and survival analysis. One hundred 3rd instar nymphs were treated for each target gene, and each treatment was carried out for three biological replicates.

To assess the adult development and fecundity, ovaries were dissected at 3 days after injection, so as to test the effect on the internal reproductive system. The dsRNA-treated females at 3 days after injection were allowed to mate with the healthy males in a glass tube for 24 h. Afterwards, the males were removed, whereas the females were allowed to oviposit for 5 days. After the hatching period, the number of hatched offspring was counted every day until no more hatching was observed. Then, the leaf sheaths of rice seedlings were dissected to count the number of eggs failing to hatch. Finally, the total number of hatched offspring and unhatched eggs in the rice seedlings was considered as the total oviposition amount. Each treatment of females was carried out for 10 biological replicates.

### 2.6. Thermal Stress Treatment after RNAi

To examine whether the inducible NlHSP70s were important to thermal tolerance or resistance, the newly emerged females were treated with dsRNA from corresponding target genes. In the fecundity assay, five treated females of each gene, after mating with wild-type males were allowed to oviposit for 2 h in the same seedlings, then the rice seedlings with laid eggs were transferred into the 37 °C environment for 24 h, and the seedlings were later moved back to the 27 °C environment for growth and hatching. The fecundity assay was carried out for 10 biological replicates.

## 3. Results

### 3.1. Identification and Characterization of the NlHSP70 and DNAJ Family Members in BPH

A total of nine HSP70 and 31 DNAJ genes were identified from the BPH genome and transcriptome databases. To more visibly present the results, the nomenclature of the DNAJ family followed the guidelines of Kampinga et al. [4], while that of the HSP70 family referred to that of the *D. melanogaster* HSP70 genes retrieved from FlyBase. Confirmed by the NCBI BLAST (https://blast.ncbi.nlm.nih.gov/Blast.cgi (accessed on 5 September 2020)), Pfam (http://pfam.xfam.org/ version 33.1 (accessed on 5 September 2020)) and SMART (http://smart.embl-heidelberg.de/ (accessed on 5 September 2020)) programs, all HSP70s contained an HSP70 conserved domain, whereas all the DNAJ sequences included a DNAJ domain. In addition, the DNAJ family was further divided into three subfamilies, namely, DNAJA, DNAJB, and DNAJC, according to the conserved domain and structural analysis (Table 1).

Gene structure was plotted by using the GSDS 2.0 online tool by aligning the genomic DNA with the full-length cDNA of each target gene. Surprisingly, among HSP70 and DNAJ genes in all the candidate genes identified from BPH, the exon–intron organization exhibited great diversity, even within the same subfamily. As shown in Figure 1A, the number of introns in NlHSP70 genes ranged from 0 to 18. Only three genes (*NlHSP70A*, *NlHSP70B,* and *NlHSP68*) had less than three introns. In the NlDNAJ family, the number of exons varied from one to 38. To be specific, there were four genes possessing only one exon (12.9%), six with 1–3 exons (19.35%), 15 (almost 1/2 of the family members, 48.39%) with 4–8 exons, and six with over 10 exons (Figure 1B). A full amino acid alignment of NlHSP70 proteins, in Figure 1C, showed that NlHSP70 family members were highly conserved with three conserved regions (GIDLGTTYS, IFDLGGGTFDVSIL, and VGGSTRIPKVQ) defining HSP70 family signatures.

More details regarding the physiological and biochemical properties of NlHSP70 and NlDNAJ genes in BPH, including gene names, protein lengths, conserved domains, MWs, isoelectric points, and subcellular localizations, are listed in Table 1. In brief, NlHSP70 genes exhibited similar physiological and biochemical properties in terms of protein lengths. The NlHSP70 family contains seven cytosolic members, one homolog in ER, and one in mitochondria.

However, the properties were quite different among the NlDNAJ family members. In addition to the common DNAJ domain, the NlDNAJC group possessed diverse domains in different regions. The majority of NlDNAJs were predicted to be localized in the nucleus, four in cytoplasm and three in mitochondrion, but none in ER.

### 3.2. Phylogenetic Relationships of N. lugens HSP Genes

To evaluate the evolutionary relationships of the two HSP families, we performed phylogenetic analysis of each family. Due to the large number in the DNAJ family, especially for the DNAJC class, all the family members were divided into two groups, namely, DNAJA, B, and DNAJC.

As observed from the phylogenetic tree in Figure 2, HSP70 genes were clustered into three major monophyletic groups, according to their predicted subcellular localizations (including cytoplasm, ER, and mitochondrion) among the different insect species. It was suggested by the phylogenetic analysis that, NlHSP70A, NlHSP70B, and NlHSP68 were clustered with inducible HSP70s from other insects, indicating that they might be heat inducible. In addition, NlHSC70 and NlHSC70-6 were clustered into distinct branches, even though the protein names were confused, all of them belonged to the HSPH/HSP110 subfamily of the HSP70 family.

The phylogenetic analysis of the DNAJ subfamilies is illustrated in Figure 3. It was observed that each member of the three classes from diverse species formed a distinct clade on the tree. However, the bootstrap values were not very high, most of which were lower than 50%, suggesting the high diversity of the DNAJ family in evolution, and the probable versatility in function.

### 3.3. Expression Patterns of NlHSP Genes in Different Tissues

Gene expression patterns can provide important clues of gene function. As a result, this study investigated the expression level of each gene by RT-qPCR. First, we determined the tissue-specific expression patterns of NlDNAJ and NlHSP70 genes. As shown in Figure 4A,B, most NlHSP70 and NlDNAJ genes had relatively high expression levels in the testis or ovary. More exactly, three genes in the HSP70 family, including *NlHSC70*, *NlHSC70-3,* and *NlHSC70-4*, showed relatively high expression levels in ovary. *NlHSP68* and *NlHSP70B* were specifically expressed in ovary and testis, respectively. Meanwhile, *NlHSC70-5* and *NlHSC70-6* showed higher expression levels both in testis and ovary, whereas *NlHSP70A* had a higher transcript level in ovary and gut. Interestingly, we found that *NlHSC70-2* showed a rather different expression pattern, with the highest transcript level in integument, followed by those in testis and ovary.

RT-qPCR analysis results of the NlDNAJ family mainly revealed two distinct expression patterns from two clusters of NlDNAJ genes in the six tissues (Figure 4B), except for *NlDNAJC12* whose transcript level was comparatively high in wing, and *NlDNAJC22* that was expressed in numerous tissues. Cluster I genes were highly expressed in ovary, including *NlDNAJC2*, *NlDNAJC10*, *NlDNAJC13*, *NlDNAJC16*, *NlDNAJC23*, and *NlDNAJC14*. All the other 23 genes (74.2% of NlDNAJs) belonged to Cluster II, and their expression levels were relatively higher in testis. However, it should be noted that apart from five genes (*NlDNAJB3*, *NlDNAJB6*, *NlDNAJC28*, *NlDNAJC31*, and *NlDNAJC32*) specifically expressed in testis, the transcript levels of the remaining NlDNAJs in Cluster II were also relatively higher in ovary. Taken together, the high expression levels in testis or ovary indicated that NlDNAJs and NlHSP70 might play important roles in the development and function of the reproductive system.

### 3.4. Expression Profiles of NlHSP Genes in Different Developmental Stages and Their Responses to Temperature Stress

Next, we explored the expression profiles across four developmental stages under both physiological (27 °C) and temperature-stressed (4 °C for cold stress and 37 °C for heat stress) conditions.

Under normal growth conditions (27 °C), five out of the nine HSP70 genes, including *NlHSC70*, *NlHSC70-2*, *NlHSC70-3*, *NlHSC70-5*, and *NlHSC70-6*, showed relatively high expression levels in adults. *NlHSC70-4* was expressed at every stage, and the highest expression level of *NlHSP70B* was in the 5th instar. For *NlHSP70A* and *NlHSP68*, their transcript levels were not high across all the tested developmental stages, but significant up-regulation was observed after heat stress treatment, especially in the adult and egg stages. Apart from *NlHSP70A* and *NlHSP68*, almost all the HSP70 genes, such as *NlHSC70*, *NlHSC70-2*, *NlHSC70-3*, *NlHSC70-4*, *NlHSC70-5,* and *NlHSP70B*, were significantly up-regulated under heat stress at the egg or adult developmental stage (Figure 4C), demonstrating their putative functions in thermal tolerance in BPH.

A majority of genes in the NlDNAJ family were highly expressed in adults under physiological conditions (27 °C). *NlDNAJB3*, *NlDNAJB6*, *NlDNAJC28*, *NlDNAJC31,* and *NlDNAJC32* were specifically expressed in adults. Moreover, *NlDNAJC8*, *NlDNAJC13,* and *NlDNAJC14* reached the highest expression levels in the egg stage. Additionally, several genes, including *NlDNAJB5*, *NlDNAJC7,* and *NlDNAJC9*, had high transcript levers both in egg and adult stages. In the case of temperature stress, most NlDNAJ genes were not obviously up-regulated. Nevertheless, a few genes were partially correlated with temperature. For example, *NlDNAJC12* seemed to be up-regulated upon cold stress in egg and nymph stages. Some genes, including *NlDNAJC10*, *NlDNAJC13*, and *NlDNAJC14*, were sensitive to low temperature in the 5th instar. *NlDNAJA1*, *NlDNAJB5,* and *NlDNAJC8* were clustered because their expression levels were higher at 37 °C compared with those at 27 °C, especially in the egg and adult stages (Figure 4D).

### 3.5. RNAi Screening Identified Several Essential HSPs in BPH

The RNAi technique has developed into a mature approach for genome-wide or target reverse genetic screening, so as to identify genes involved in various cellular pathways. In addition, BPH has been demonstrated as a perfect model, with high RNAi efficiency. Consequently, RNAi screening was carried out in this study to identify NlHSPs with essential functions. Finally, seven indispensable HSP70s and 10 DNAJs in BPH were identified among the 40 HSP genes.

#### 3.5.1. Indispensable Roles of HSP70 Genes in BPH Viability and Female Reproduction under Normal Conditions

To investigate the potential function of NlHSP70s in nymphal development, the RNAi technique was applied to the 3rd instar nymphs. As a result, RNAi against four HSP70 genes (*NlHSC70*, *NlHSC70-3*, *NlHSC70-4*, and *NlHSC70-5*) led to lethal phenotypes with different mortality. Among these four genes, knockdown of *NlHSC70-3*, *NlHSC70-4,* and *NlHSC70-5* resulted in a relatively higher mortality rate, with the survival rate of <20% at eight days after injection (Figure 5A). The lethal effect of *NlHSC70* injection was not as obvious as those of three NlHSCs, which demonstrated the survival rate of 69.35% (Figure 5B). Intriguingly, the lethal phenotypes were quite similar after injection of ds*NlHSC70-3*, ds*NlHSC70-4,* and ds*NlHSC70-5*. As shown in Figure 5C, nymphs treated with dsRNA of the three target genes died before, or in, the ecdysis stage, while the ds*NlHSC70*-treated nymphs had difficulties in adult emergence, causing lethality in the late stage. In addition, the surviving adults showed abnormal wing morphosis (Figure 5D). At 3 days after adult emergence, the insects were dissected, which revealed that the ovary was malformed with round oocytes (Figure 6A).

It was discovered from the RT-qPCR results that most NlHSP70 genes had relatively high expression levels in ovary. Therefore, we further performed an RNAi experiment on the newly emerged females to investigate the impacts of these genes on female fecundity. The ovary phenotype was observed, which suggested that three genes, namely, *NlHSC70-3*, *NlHSC70-4,* and *NlHSC70-5*, led to severe ovary malformation compared with the banana shape of the control female ovaries. As is also illustrated in Figure 6A, oocytes were dysplastic and there seemed to be no vitellogenin accumulated in the oocytes. As a result, female adults treated with *NlHSC70-3*, *NlHSC70-4,* and *NlHSC70-5* were unable to produce any eggs. Although no obvious abnormality was observed in the ovaries after knockdown of *NlHSC70-2*, *NlHSP68,* and *NlHSP70A*, the reproductive ability (manifested as total number of eggs and hatchability) was affected. The average egg numbers produced by females treated with ds*NlHSP70A* and ds*NlHSC70-2* were 88 and 8.5, respectively, which was significantly decreased compared with that produced by ds*GFP*-treated females (162 eggs per female) (Figure 7A). Accordingly, the hatchability was 57.3% under ds*NlHSP68* treatment, while almost no offspring hatched in the ds*NlHSP70A* and ds*NlHSC70-2* groups (Figure 7B).

Based on the similar phenotypes observed among *NlHSC70-3*, *NlHSC70-4,* and *NlHSC70-5*, we further analyzed the effect of silencing one gene on the other two genes. RT-qPCR results showed that when knocking down *NlHSC70-4* in the 3rd instar nymphs, the relative expression of *NlHSC70-5* was significantly down-regulated at 3 days after injection (Figure 8). There were no significant difference in other groups.

#### 3.5.2. Thermal Tolerance and Biological Functions of Two High Temperature Inducible HSP70 Genes

As confirmed by the RT-qPCR results, two genes (*NlHSP70A* and *NlHSP68*) were obviously up-regulated in various developmental stages upon high temperature stress, which prompted us to further investigate the thermal tolerance and gene functions of these two genes under heat stress.

Meanwhile, the oviposition and egg hatching trial under normal growth condition (27 °C) showed that knockdown of *NlHSP70A* led to the lower egg production, whereas *NlHSP68* had no distinct effect on oviposition or egg amount. However, the hatchability was affected at different levels. Specifically, *NlHSP70A* silencing led to hatching failure, and the hatching rate decreased by nearly a half after *NlHSP68* RNAi. Furthermore, some eggs laid by ds*NlHSP70A*-treated females were dry in the seedling, whereas the rest stopped embryonic development before eye spot formation (Figure 6C). Considering that the loss-of-function of *NlHSP70A* caused severe effects on egg production and embryonic development under the normal condition, we focused on *NlHSP68* upon high temperature stress. Interestingly, after 24 h of heat stress treatment, no eggs hatched successfully after ds*NlHSP68* treatment (Figure 7C). Next, we dissected the seedling on day 8, a time point when the control eggs developed red eye spots and clear appendages. As a result, the eggs laid by the treated females were undeveloped, most of which seemed dry and wizened (Figure 6C). Together, these results indicated the important role of *NlHSP68* in the thermal tolerance of egg protection.

#### 3.5.3. Essential Roles of NlDNAJs in BPH Development

The NlDNAJ family comprises the largerfamily members. First of all, each of the 31 NlDNAJ dsRNAs was injected into the thorax of the 3rd instar nymphs. In general, a total of seven genes (including 2 belonging to the DNAJA subfamily and 5 belonging to the DNAJC subfamily) led to lethality, but with different phenotypes, which were classified as three classes. Intriguingly, injection with ds*NlDNAJA3* and ds*NlDNAJC19* resulted in a similar Phenotype I, where the difficulty in ecdysis was also observed together with burned-like black on the edge of the insects, e.g., the antenna or spur in the 5th instar nymphs (Figure 9C). The treated BPH nymphs of Phenotype II died before, or in, the ecdysis stage (Figure 9D). Moreover, RNAi against *NlDNAJC5* and *NlDNAJC22* resulted in a high mortality rate, and almost all the treated nymphs were unable to develop into the next instar successfully. The rapid response and high mortality rate indicated that this protein plays an essential role in the development and lifespan of insects. *NlDNAJC13* also induced ecdysis difficulty, with a relatively low mortality rate (about 30% in Figure 9D). The treated BPHs of Phenotype III had molting difficulties or abnormal organs after adult emergence. *NlDNAJA1*, *NlDNAJC2,* and *NlDNAJC16* were classified into this group. In detail, injection with ds*NlDNAJC16* resulted in molting failure, and most nymphs died during the molting process. Moreover, *NlDNAJA1* silencing also caused molting failure, with abnormal wing blistering after adult emergence (Figure 9E). No significant mortality was observed after RNAi against *NlDNAJC2*, but ovarian malformation was seen (Figure 6B).

#### 3.5.4. Essential Roles of NlDNAJs in Female Fecundity

It has been demonstrated that parental RNAi can be delivered to the next generation of BPHs. As a result, this study conducted RNAi experiments on the newly emerged female adults before the maturation of their ovaries, and measured the fecundity by means of ovary observation, egg production, and hatchability.

Interestingly, although a number of DNAJ genes showed high expression levels in ovary, only treatment with ds*NlDNAJC5* and ds*NlDNAJC17* led to abnormalities in oocytes (Figure 6B). Furthermore, it was shown by oviposition trial that there were four genes related to egg production or hatchability. Females experiencing RNAi against *NlDNAJA3* and *NlDNAJC13* barely laid eggs, and none of them hatched successfully (Figure 7D,E). Knockdown of *NlDNAJA1* had no influence on oviposition, but it severely affected embryogenesis. As shown in Figure 6C, development arrest was detected in the eggs on day 8. The average egg production number of *NlDNAJC14*-treated females was 108, which was significantly lower than that of the GFP group. In addition, the hatchability significantly decreased to 52.0% after *NlDNAJC14* treatment (Figure 7D,E).

## 4. Discussion

HSP70s and DNAJs are the ubiquitous molecular chaperones that function in a myriad of biological processes under both physiological and stressful conditions [1,8]. On the one hand, the sequences and functions of HSP70s are highly conserved across different species; on the other hand, the great diversity of DNAJs endows the HSP70/DNAJ machinery with the potential for multifunctional and client specificities [8,26]. Consequently, despite the progresses achieved in HSP70s and DNAJs research, more studies are warranted to investigate the expression profiles and biological functions of HSP70 and DNAJ family genes, let alone the big gap of knowledge in insects.

In this study, we identified and characterized a total of nine HSP70 genes and 31 DNAJ genes of *N. lugens*. The quantity distributions of these two families in BPH are consistent with those in other species, with the number of DNAJs being greater than that of HSP70s generally [8,11]. Furthermore, according to gene structure and phylogenetic analysis, these two families can be divided into distinct classes. For BPH, there is one HSP70 homolog in mitochondria and one in ER, whereas the rest are localized in cytoplasm. *NlHSC70* and *NlHSC70-6* may be clustered in the HSPH/HSP100 subfamily. With the assistance of RT-qPCR, we investigated the expression patterns of all genes in different tissues and developmental stages, both under normal and temperature-stressed conditions. In particular, we identified that seven NlHSP70 genes and 10 NlDNAJ genes played indispensable roles in BPH viability and demonstrated that NlHSP68 was important to the embryo thermal tolerance by RNAi. Recently, although it was demonstrated in some studies that Hsc70/Hsp90 chaperone machinery participates in RNAi pathways [27], RNAi has been used as a sufficient tool in many species to explore the in vivo function of HSPs. Our RNA-seq after RNAi of two HSP90 homologs (NlHsp90 and NlGRP94) in female BPH did not show an interaction between HSP90 and RNAi pathways (unpublished data).

The advances in genome sequencing have largely contributed to the identification of HSP70 homologs from insects classified in diverse orders [2,28,29,30,31]. Previously, one inducible Hsp70 and one heat shock cognate protein 70 (Hsc70) were cloned from *N. lugens*, and their expression patterns in adult females were analyzed across temperature gradients [22,23]. As far as we know, this study was the first to systematically perform a genome-wide identification of HSP70 genes in BPH. In addition, RT-qPCR results under temperature stress indicated that almost all the HSP70 family members were up-regulated in egg and adult stages, while these two periods were vital for reproduction, indicating the potential roles of NlHSP70s in thermal resistance. Nevertheless, this study also identified that two inducible cytoplasmic HSP70s (*NlHSP70A* and *NlHSP68*) were most significantly regulated by high temperature in all the developmental stages.

Due to the typical function of HSP70s in temperature stress resistance, most studies have focused on the expression pattern and survival analysis under extreme environmental stress [2,29,30,31,32]. However, only a few studies have demonstrated the essential roles of HSP70s under normal condition. For example, the *D. melanogaster* HSC70s (*HSC3* and *HSC4*) are required for proper tissue establishment and maintenance, while mutations in these two genes will lead to lethality at normal temperatures [33]. In the present study, we focused on the potential gene functions not only at physiological condition but also under temperature stress. As revealed by the RNAi-mediated loss-of-function, ablation of seven out of the 9 NlHSP70 genes gave rise to abnormalities at different levels, which revealed the indispensable roles of HSP70 genes in BPH. In addition, the knockdown of four HSP70 genes (*NlHSC70*, *NlHSC70-3*, *NlHSC70-4,* and *NlHSC70-5*) induced lethal phenotypes, indicating that these genes might exert essential biological functions in nymph development under physiological condition. It has been demonstrated that HSP70s are implicated in female fertility and healthy egg production. For instance, it was previously demonstrated that *Drosophila hsp70-4* (*NlHSC70-4* homolog) is an important contributor to the production of a healthy egg [34]; in addition, during a screening for genes involved in regulating transposon silencing in oocytes, *DmHsp70-5* was found among the 100 strongest hits, but further information about its role is lacking [35]. Nonetheless, we still found three interesting discoveries based on our phenotype observation.

First, knockdown of three genes (*NlHSC70-3*, *NlHSC70-4,* and *NlHSC70-5*) exhibited similar phenotype (to be specific, higher mortality in treated nymphs, and ovary maldevelopment in the treated newly emerged females. Intriguingly, the three HSP70 genes were predicted to be localized in the distinct subcellular compartments, including cytoplasm, mitochondria, and ER. The sequence regions designed for dsRNA synthesis were not homologous with each other. Additionally, we further tested the relative expression levels of two HSP70 genes when the other one was knocked down. As a result, the expression level of *NlHSC70-5* was significantly down-regulated only when *NlHSC70-4* was silenced, indicating a possible interaction of *NlHSC70-4* with *NlHSC70-5* at RNA level. However, more studies are needed to investigate whether the three HSP70 genes interact with each other at a protein level, and how they interact with each other in different subcellular compartments. Second, our results suggested that *NlHSC70-2*, a gene highly expressed in integument, might function in egg formation and development, because of the low egg production and hatchability. As revealed by Blast analysis, our *NlHSC70-2* was identical to the *NlHsp70* reported by Lu and colleagues, which exhibited similar highest expression levels in the epidermis [23]. In this regard, our results revealed a novel biological function of this gene under normal conditions. Third, we demonstrated that two inducible HSP70 genes were involved in the embryogenesis and thermal tolerance of eggs laid in the leaf sheaths of rice seedling. RNAi against *NlHSP70A* caused hatching failure under normal growth conditions, and dried and undeveloped eggs were obtained. A total of 57.3% of the laid eggs hatched successfully when treated with *NlHSP68* under physiological conditions, while no egg hatched under the 24-h heat stress, suggesting that *NlHSP68* played an important role in the thermal tolerance of eggs. Meanwhile, research on *Agasicles hygrophila* examined the relationships between the transcript levels of 26 HSP genes and the heat tolerance of eggs, but more information is needed [36]. Our results provided strong evidence supporting the role of *NlHSP68* in the thermal tolerance of eggs.

Generally, it has been assumed that HSP70 paralogs have similar activities and are largely functionally redundant. However, the accumulated studies have revealed the diversity of HSP70s. It has been demonstrated that different HSP70 isoforms have unique cochaperone and client interactors, and functional diversity is defined by the affinities for specific cochaperones [37,38,39]. Together, our discoveries may offer novel clues to explore the relationships between HSP70 family members and uncover the underlying mechanism of HSP70s in contributing to oogenesis, embryogenesis, and the thermal tolerance of eggs.

The DNAJ family was classified into three subgroups based on the domain structure. Among them, the DNAJC class (23) occupied most of the family members, while five were in DNAJB class, and three were in the DNAJA class. Furthermore, gene architecture and phylogenetic analysis revealed the large diversity of DNAJ genes, especially for the DNAJC class, indicating that this family might have versatile functions. Moreover, tissue expression patterns of DNAJ genes underscored the importance of a set of chaperones in the development of the reproductive system and the function of BPH. Transcriptome and proteome analysis in *Caenorhabditis elegans* also identified a distinct series of heat shock-related genes (including HSP70 and DNAJ genes) expressed at detectable levels in gonads, providing clues of the functions of HSP genes in oogenesis or spermatogenesis [40,41,42]. However, different mutants and technologies have been exploited in diverse experiments, which may cause great differences in bioinformatics analysis results. Consequently, we applied RNAi knockdown in further validating the biological functions of DNAJ genes.

Unexpectedly, although a certain number of DNAJ genes (including five testis-specific genes) had relatively high expression levels in testis, no obvious phenotype was observed after RNAi in male BPHs, including the development of male reproductive system and fertility. More than one DNAJ protein functions together with each HSP70, which prompts us to speculate that there may be complementary relationships between different DNAJ members. Previous studies have demonstrated that several DNAJ proteins exist and function in the male germline. For example, *DjA1* (type I DnaJ homolog)-mutant mice led to severe defects in spermatogenesis via androgen receptor-mediated signaling [43]. However, it seemed that knockdown of *NlDNAJA1* did not affect spermatogenesis; instead, it severely affected nymph development, adult wing morphology, and embryogenesis (Figure 6C and Figure 9). This observation might expand the functional role of *DNAJA1*.

Two mitochondrial members, *NlDNAJA3* and *NlDNAJC19*, exhibited similar phenotype in nymphs after gene knockdown. Apart from ecdysis difficulty, burned-like black was also observed on the edge of insects, on the antenna, wing, or spur in the 5th instar nymphs. *Drosophila* Tid56, the orthologue of *NlDNAJA3*, was first discovered as a tumor suppressor, and its loss-of-function will cause malignant growth of imaginal discs cells and lethality of the mutant larvae at the time of puparium formation [44]. Moreover, *TID1*, a mammalian homologue of *Tid56*, is proposed to be critical for early embryonic development and cell survival [45]. Our results showed that *NlDNAJA3* was also essential for nymph development, and RNAi-treated females hardly laid eggs, revealing that this gene had a conserved function across different species. However, the burned-like and tissue necrosis phenotype has not been reported, which may suggest the novel function of *NlDNAJA3*. On the other hand, *NlDNAJC19* is homologous to *Tim14* in yeast. Anchored in the inner mitochondrial membrane, *Tim14* is a constituent of the mitochondrial import motor, which stimulates the ATPase activity of mtHsp70 [46,47]. In yeast, the loss-of-function of the J-domain in Tim14 is lethal, but there have been no studies on this gene in insects. This study first investigated the potential role of *NlDNAJC19* in the nymph development of BPH. The similar phenotype found in nymphs after RNAi revealed certain links between the two mitochondrial NlDNAJs. Andy Cheuk-Him Ng. et al. showed that *TID1* was essential in maintaining the homogeneity of mitochondrial membrane potential (MMP) [48]. As a result, it was hypothesized that preproteins may be transported across the inner mitochondrial membrane with the aid of *NlDNAJC19*, which then interacts with *NlDNAJA3*. Nevertheless, their distinct functions in females suggested that they might be involved in other diverse pathways, because the knockdown of *NlDNAJC19* in females had no influence, but ds*NlDNAJA3* treatment led to oviposition failure.

Apart from the above-mentioned DNAJ genes, there are several other genes functioning in various aspects. According to our results, when nymphs were treated with corresponding dsRNA, *NlDNAJC5*, *NlDNAJC13,* and *NlDNAJ22* led to lethality before or during ecdysis, indicating that these genes might play indispensable roles in the development of BPH. In addition, *NlDNAJC16* caused molting failure. Injection of ds*NlDNAJC2* in the 3rd instar nymphs severely affected the development of the ovary. Furthermore, the female fecundity trial revealed that *NlDNAJC13* was an important contributor to the production of healthy eggs, whereas *NlDNAJC5* and *NlDNAJC17* played vital roles in oocyte development. Taken together, DNAJ genes exhibited functional versatility consistent with their structural diversity.

It is already known that HSP70s function with their cochaperone DNAJs. However, our results showed that the functions of most DNAJs were partially rather than entirely consistent with those of HSP70s. For example, silencing of *NlDNAJC5* was the closest to *NlHSC70-3*, *NlHSC70-4,* and *NlHSC70-5,* both in the treated nymphs and females, indicating the potential interactions between them. Furthermore, research has demonstrated that Tim14 and TID interact with mtHSP70 [45,46]. However, in our study, two homologs, *NlDNAJA3* and *NlDNAJC19,* did not show exactly the same phenotype with NlHSC70-5, the mitochondrial member. Apart from nymph survival, *NlHSC70-5* severely defected ovary development, whereas the two DNAJs made no difference to ovary, and *NlDNAJA3* was involved in egg production. Injection of both *NlDNAJC2* and *NlHSC70* induced ovary malformation in the treated nymphs, but *NlHSC70* led to lethality in the molting stage, while *NlDNAJC2* did not. Another gene, *NlDNAJC16*, also affected the molting process but had no effect on ovary. More experiments at a protein level should be conducted to confirm whether *NlDNAJC2* together with *NlDNAJC16* interacts with *NlHSC70*. Consequently, though DNAJs play roles of cochaperones to their HSP70 partners, versatile biological roles independent of HSP70 were discovered in our study. More recently, accumulated evidence has uncovered a vast array of post-translational modifications on Hsp70 family proteins, including phosphorylation, methylation, acetylation, ubiquitination, AMPylation, and ADP-ribosylation [49]. As a result, differences in function of the HSP70/DNAJ machinery might arise from expression and sequence differences, as well as differential post-translational modification of HSP70 isoforms.

## 5. Conclusions

In conclusion, this study was the first to systematically identify and characterize the HSP70 and DNAJ family members in BPH. According to the RNAi-mediated loss-of-function analysis, seven HSP70s and 10 DNAJs play vital roles in nymph development, oogenesis, and female fertility of *N. lugens* under physiological growth conditions; additionally, one HSP70 is important in the thermal tolerance of eggs. Together, our findings shed more light on the HSP70/DNAJ machinery, and provide clues for the interactions within or between the two families, and lays certain foundations for exploring the specific function of DNAJs independent of HSP70s.

## Figures and Tables

**Figure 1 genes-12-00394-f001:**
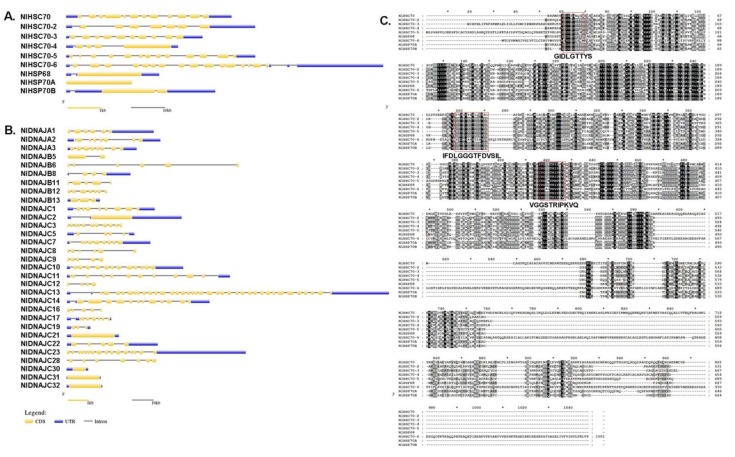
Exon–intron organization of NlHSP70 (**A**) and NlDANJ (**B**) genes. The yellow box, blue box, and gray line represent the coding sequence (CDS), untranslated region, and intron, respectively. (**C**) A full amino acid sequence alignment of NlHSP70 proteins. Identical amino acid residues and conservative substitutions are shaded in black and gray, respectively. HSP70 family signatures are indicated with red boxes.

**Figure 2 genes-12-00394-f002:**
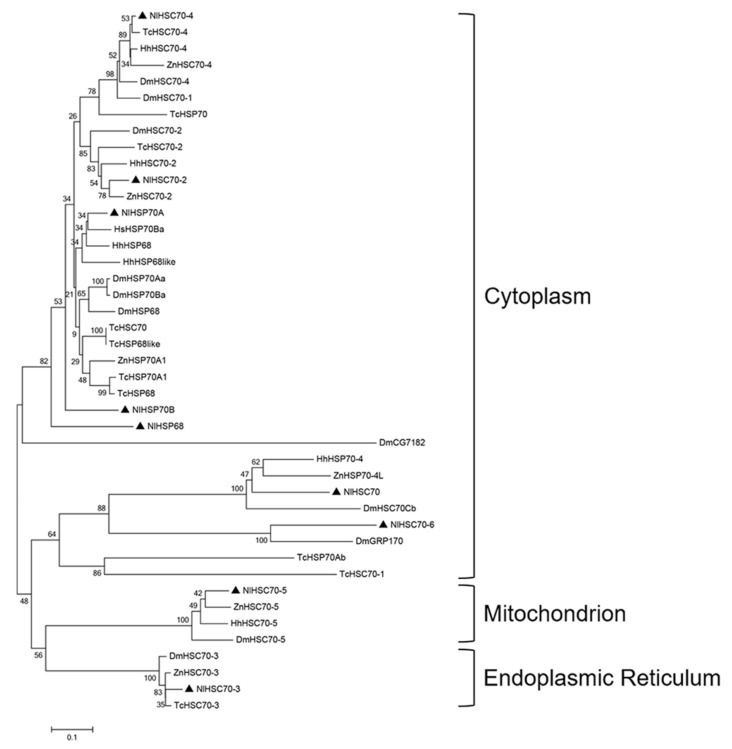
Phylogenetic analysis of HSP70 genes in BPH. The phylogenetic tree was constructed based on 47 HSP70 homologs derived from five species (Appendix A) by the neighbor-joining method. Bootstraps were set with 1000 replications. NlHSP70 members are shown in black triangles.

**Figure 3 genes-12-00394-f003:**
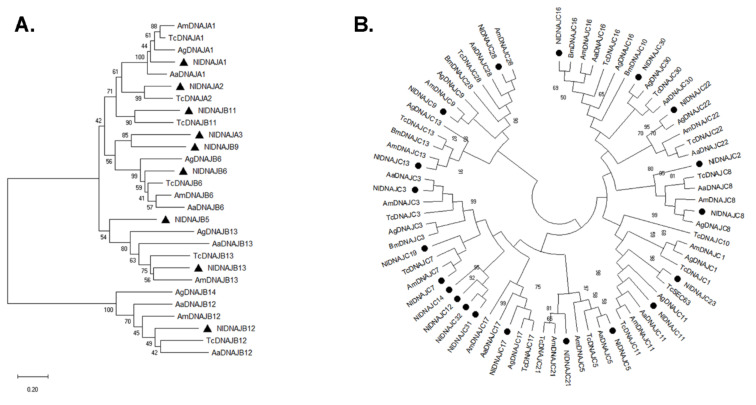
Phylogenetic analysis of DNAJ genes in BPH. (**A**) The phylogenetic tree constructed based on 25 DNAJA and DNAJB homologs derived from four species (Appendix A). (**B**) The phylogenetic tree constructed based on 51 DNAJC homologs derived from five species (Appendix A). The phylogenetic trees were constructed by the neighbor-joining method and bootstraps were set with 1000 replications. NlDNAJ members are shown with black circles.

**Figure 4 genes-12-00394-f004:**
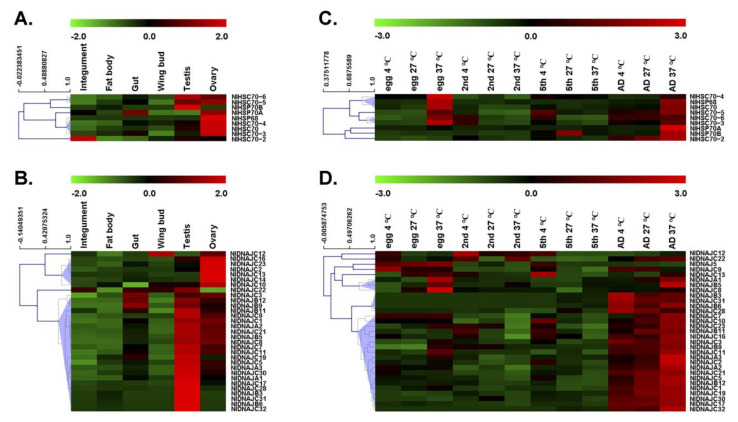
Spatio-temporal expression patterns of NlHSP70 and NlDNAJ genes. (**A**,**C**) Tissue expression patterns of NlHSP70 and NlDNAJ genes. Total RNAs were isolated from the integument, fat body, gut, ovary, and testis of BPH. (**B**,**D**) Developmental profiles of NlHSP70 and NlDNAJ genes at different temperatures. Total RNAs were isolated from eggs, 2nd instar, 5th instar nymphs, and adults at 24 h after molting under 4 °C, 27 °C, and 37 °C for 1 h. RT-qPCR and the ΔΔCt method were utilized to measure the relative transcript levels. The heatmap was created using the MultiExperiment Viewer (MeV) software.

**Figure 5 genes-12-00394-f005:**
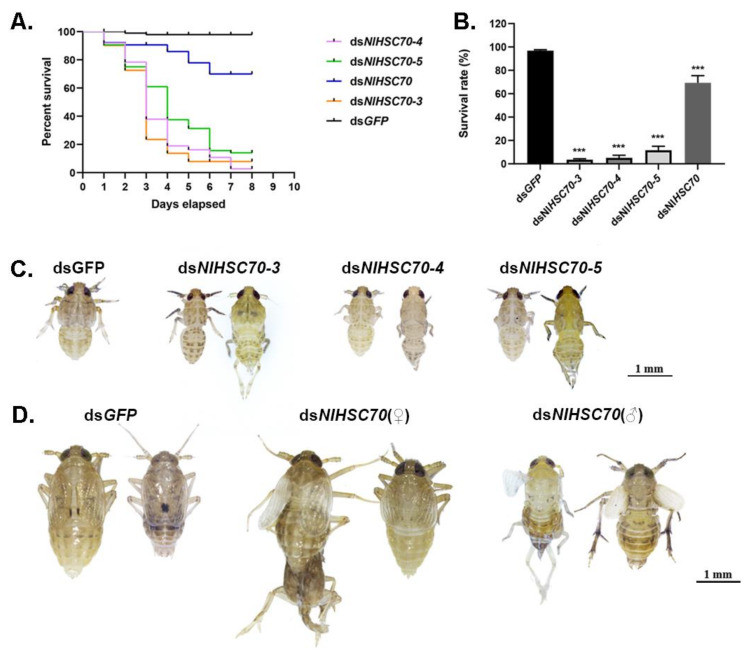
Effects of ds*NlHSP70*s treatments on nymphs. (**A**) The survival curve of BPH at 8 days after knockdown when the 3rd instar nymphs were injected with ds*NlHSP70*s. (**B**) The survival rate of BPH on day 8 after knockdown. Results are presented as mean ± SEM. *** *p* < 0.001 (Student *t*-test). (**C**) Effects of ds*NlHSC70-3*, ds*NlHSC70-4,* and ds*NlHSC70-5* on the development of nymphs. (**D**) The phenotype of nymphs after injection with ds*NlHSC70* during molting.

**Figure 6 genes-12-00394-f006:**
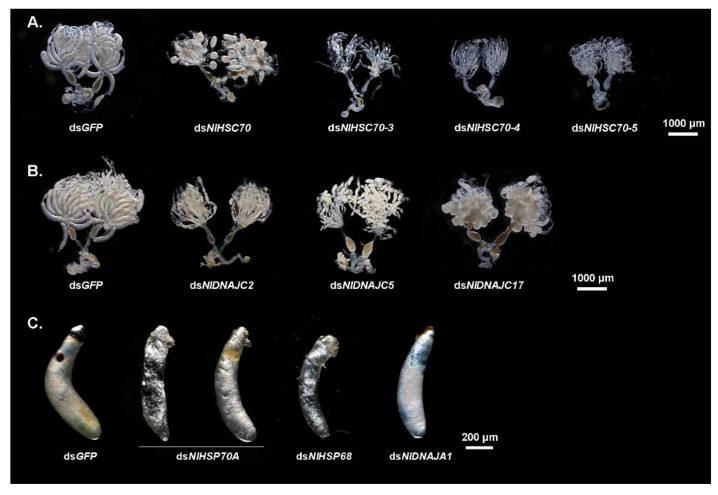
Effects of dsRNA treatments on ovary and embryonic development. (**A**) Ovaries dissected from ds*NlHSC70*-, ds*NlHSC70-3*-, ds*NlHSC70-4*-, and ds*NlHSC70-5*-treated females at 3 days after adult eclosion. (**B**) Ovaries dissected from ds*NlDNAJC2*-, ds*NlDNAJC5*-, and ds*NlDNAJC17*-treated females at 3 days after adult eclosion. ds*GFP* was used as the negative control. (**C**) Embryonic development after injection with dsRNA on day 5. The eggs laid by ds*GFP*-treated females served as the parallel controls.

**Figure 7 genes-12-00394-f007:**
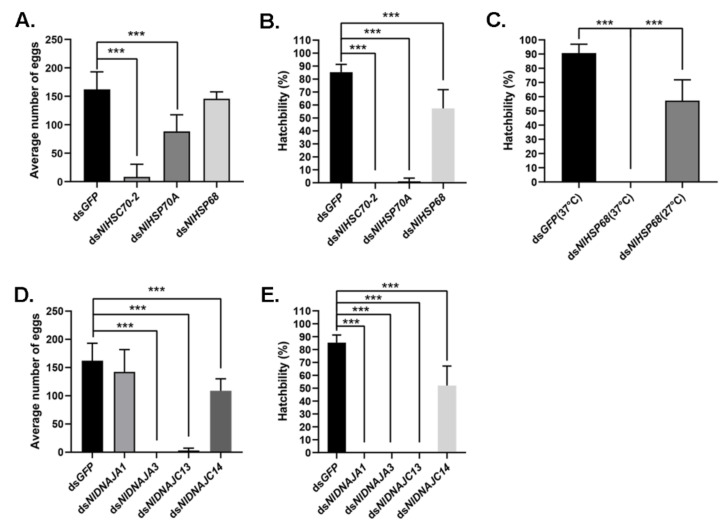
Effects of dsRNA treatments on female fecundity. (**A**,**D**) Average number of eggs laid by each female after different RNAi treatments. (**B**,**E**) Egg hatchability after RNAi treatment under normal conditions. (**C**) Egg hatchability of *NlHSP68* after RNAi treatment under normal conditions and thermal stress. For each treatment, 10 replicates were set. Results are presented as mean ± SEM. *** *p* < 0.001 (Student *t*-test).

**Figure 8 genes-12-00394-f008:**
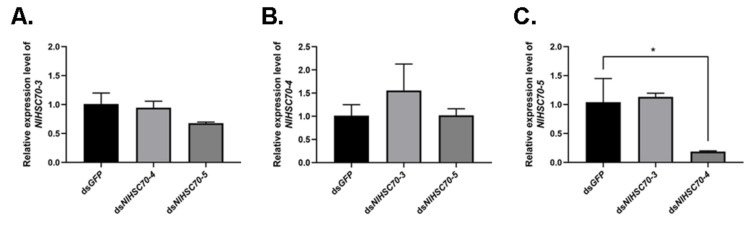
(**A**) The relative expression of *NlHSC70-3* when *NlHSC70-4* and *NlHSC70-5* were knocked down. (**B**) The relative expression of *NlHSC70-4* when *NlHSC70-3* and *NlHSC70-5* were knocked down. (**C**) The relative expression of *NlHSC70-5* when *NlHSC70-3* and *NlHSC70-4* were knocked down. ds*GFP* was used as the negative control. Results are presented as mean ± SEM. * *p* < 0.05 (Student *t*-test).

**Figure 9 genes-12-00394-f009:**
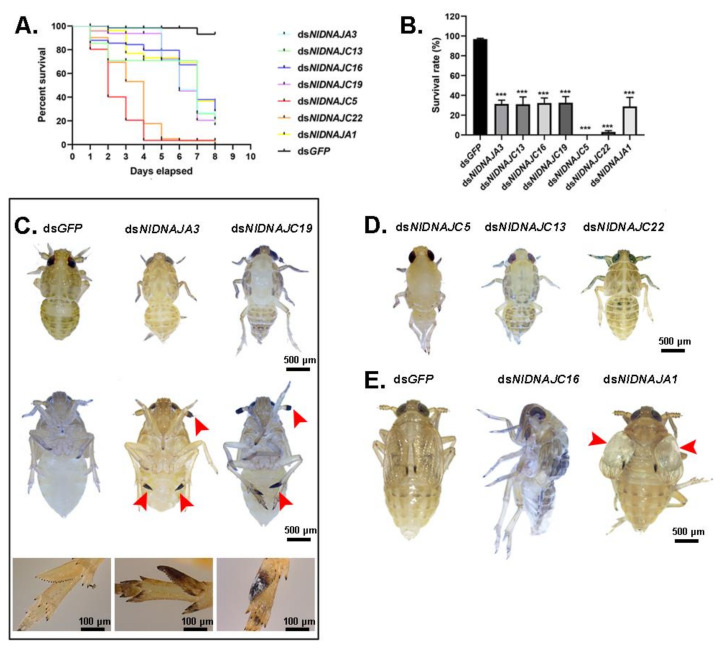
Effects of ds*NlDNAJ*s treatments on BPH. (**A**) The survival curve of BPH at 8 days after knockdown when the 3^rd^ instar nymphs were injected with ds*NlDNAJs*. (**B**) The survival rate of BPH on day 8 after knockdown. Results are presented as mean ± SEM. *** *p* < 0.001 (Student *t*-test). (**C**) Effects of ds*NlDNAJA3* and ds*NlDNAJC19* on the development of nymphs (Phenotype I). The burned-like black on antenna or spur is indicated by red arrow heads. (**D**) Phenotypes of nymphs after injection with ds*NlDNAJC5*, ds*NlDNAJC13,* and ds*NlDNAJC22* during ecdysis (Phenotype II). (**E**) Phenotypes of nymphs after injection with ds*NlDNAJC16* and ds*NlDNAJA1* during molting (Phenotype III). ds*GFP* was injected as a negative control.

**Table 1 genes-12-00394-t001:** Summary of DNAJ and HSP70 genes in brown planthopper (BPH).

Gene Name	Protein Length (aa)	Domain Structure	Localization	PI	Mw (kDa)	GenBank Accession Number
*NlDNAJA1*	403	DnaJ-CXXCXGXG	Nuclear	6.54	45,207.75	XM_022336997.1
*NlDNAJA2*	404	DnaJ-CXXCXGXG	Nuclear	5.93	45,206.38	XM_022341078.1
*NlDNAJA3*	517	DnaJ-CXXCXGXG	Mitochondrial	9.06	57,148.44	XM_022340644.1
*NlDNAJB5*	362	DnaJ-DnaJ_C	Cytoplasmic	9.19	40,027.36	XM_022340289.1
*NlDNAJB6*	280	DnaJ	Cytoplasmic	8.79	31,411.26	XM_022342229.1
*NlDNAJB9*	203	DnaJ	Extracellular	5.81	23,776.23	XM_022329969.1
*NlDNAJB11*	348	DnaJ-DnaJ_C	Nuclear	5.87	40,765.03	XM_022328246.1
*NlDNAJB12*	367	DnaJ-DUF1977	Nuclear	9.02	42,611.88	XM_022347944.1
*NlDNAJB13*	343	DnaJ-DnaJ_C	Cytoplasmic	6.18	38,975.92	XM_022345508.1
*NlDNAJC1*	404	DNAJ-Myb_DNA-binding	Nuclear	5.93	56,342.91	XM_022338476.1
*NlDNAJC2*	636	DNAJ-RAC_head-Myb_DNA-binding	Nuclear	8.80	73,871.36	XM_022332478.1
*NlDNAJC3*	492	TPR_19-TPR_2-DNAJ	Cytoplasmic	6.09	56,531.95	XM_022349039.1
*NlDNAJC5*	280	DnaJ	Extracellular	8.34	19,145.78	XM_022335030.1
*NlDNAJC7*	432	TRP_16 DNAJ	Nuclear	7.15	55,854.52	XM_022341451.1
*NlDNAJC8*	255	DnaJ	Nuclear	8.81	30,132.09	XM_022340345.1
*NlDNAJC9*	268	DnaJ	Nuclear	8.25	31,660.55	XM_022338663.1
*NlDNAJC10*	705	DNAJ-Thioredoxin-Thioredoxin-Thioredoxin-Thioredoxin	Nuclear	6.36	90,197.36	XM_022352275.1
*NlDNAJC11*	581	DnaJ-DUF3395	Nuclear	6.91	65,104.83	XM_022345278.1
*NlDNAJC12*	174	DnaJ	Nuclear	6.6	19,462.64	XM_022347075.1
*NlDNAJC13*	1935	GYF_2 DNAJ	Nuclear	6.52	254,644.11	XM_022351495.1
*NlDNAJC14*	774	DnaJ-Jiv90	Nuclear	9.71	110,737.38	XM_022337903.1
*NlDNAJC16*	226	DNAJ-Thioredoxin	Extracellular	6.38	23,740.26	XM_022344144.1
*NlDNAJC17*	301	DNAJ-RRM_1	Nuclear	6.83	34,460.93	XM_022343865.1
*NlDNAJC19*	118	transmembrane region-DnaJ	Mitochondrial	9.94	12,878.14	XM_022330675.1
*NlDNAJC21*	670	DNAJ-zf-C2H2_jaz	Nuclear	4.79	77,404.98	XM_022341932.1
*NlDNAJC22*	147	TM2-DNAJ	PlasmaMembrane	8.92	41,680.39	XM_022341798.1
*NlDNAJC23*	1224	DNAJ-Sec63-Sec63	Nuclear	5.71	90,974.22	XM_022329642.1
*NlDNAJC28*	401	DnaJ-DUF1992	Nuclear	8.98	46,881.25	XM_022345681.1
*NlDNAJC30*	203	DnaJ	Mitochondrial	9.47	23,471.83	MW551888
*NlDNAJC31*	388	DnaJ	Nuclear	8.11	57,091.78	XM_022338345.1
*NlDNAJC32*	500	DnaJ	Nuclear	8.09	56,984.84	XM_022338345.1
*NlHSC70*	825	HSP70	Cytoplasmic	5.35	91,500.7	KU932300.1
*NlHSC70-2*	631	HSP70	Cytoplasmic	5.84	68,935.24	KU932402.1
*NlHSC70-3*	667	HSP70	**Endoplasmic reticulum**, ER	5.35	74,325.34	XM_022337451.1
*NlHSC70-4*	689	HSP70	Mitochondrial	5.69	74,833.02	XM_022345583.1
*NlHSC70-5*	624	HSP70	Cytoplasmic	5.42	68,279.17	XM_022345825.1
*NlHSC70-6*	1001	HSP70	Cytoplasmic	5.31	111,495.69	KU932221.1
*NlHSP68*	627	HSP70	Cytoplasmic	5.35	91,500.7	KX976476.1
*NlHSP70A*	646	HSP70	Cytoplasmic	5.47	71,116.21	XM_022341134.1
*NlHSP70B*	624	HSP70	Cytoplasmic	5.42	68,279.17	XM_022345825.1

## Data Availability

Data are contained within the article or Appendix A.

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
