# Peer review of "HSP70/DNAJ Family of Genes in the Brown Planthopper, Nilaparvata lugens: Diversity and Function"

_genes, 2021, doi:10.3390/genes12030394_

Round 1
Reviewer 1 Report
In the manuscript “"HSP70/DNAJ family of genes in the brown planthopper, Nilaparvata lugens: diversity and function", the authors investigate a number of genes in HSP70/DNAJ family in a pest species brown planthopper. To achieve that the authors identify and characterize the members of this gene family based on genome and transcriptome data, investigate the gene expression in various tissues and perform a wide-scale RNAi screening. While I would like to acknowledge the amount of the work done by the authors, I have a major concern about the study method. The authors have chosen to study heat shock proteins (HSPs) with RNAi. The dsRNA upon its injection is processed by the RNA pathway in the cell which, according to some studies, involves Hsc70/Hsp90 chaperones machinery (see e.g. Evgen’ev, 2014). HSPs are the part of many biological processes, but alternative explanation could be that the altered expression of HSPs could “break” the RNAi pathway and result in altered expression of the other genes, altered phenotypes as well as lethality of the individuals. Though the role of HSPs in RNAi pathway is not studied well, I still consider utilizing the pathway, which involves itself the studied gene, to be prone for potential biases and resulting in a major design flaw. I wonder whether the authors have some explanation which they did not present in the Discussion.
Minor comments:
Parts 2.5 and 2.6: what are sample sizes for the treatments?
References:
- Evgen’ev, M. B., Garbuz, D. G., & Zatsepina, O. G. (2014). Heat Shock Proteins and Whole Body Adaptation to Extreme Environments. doi:10.1007/978-94-017-9235-6
Author Response
Reviewer1
Comments and Suggestions for Authors
In the manuscript “"HSP70/DNAJ family of genes in the brown planthopper, Nilaparvata lugens: diversity and function", the authors investigate a number of genes in HSP70/DNAJ family in a pest species brown planthopper. To achieve that the authors identify and characterize the members of this gene family based on genome and transcriptome data, investigate the gene expression in various tissues and perform a wide-scale RNAi screening. While I would like to acknowledge the amount of the work done by the authors, I have a major concern about the study method. The authors have chosen to study heat shock proteins (HSPs) with RNAi. The dsRNA upon its injection is processed by the RNA pathway in the cell which, according to some studies, involves Hsc70/Hsp90 chaperones machinery (see e.g. Evgen’ev, 2014). Comment: 1) HSPs are the part of many biological processes, but alternative explanation could be that the altered expression of HSPs could “break” the RNAi pathway and result in altered expression of the other genes, altered phenotypes as well as lethality of the individuals. Though the role of HSPs in RNAi pathway is not studied well, I still consider utilizing the pathway, which involves itself the studied gene, to be prone for potential biases and resulting in a major design flaw. I wonder whether the authors have some explanation which they did not present in the Discussion.
Response: Thank you for positive comments. Regarding your concern of the RNAi used for characterization of HSPs in this study, we have some explanation: Genes involved in the BPH RNAi pathway (siRNA and miRNA pathways) have been genome-widely identified and comprehensively investigated in our lab (Xu et al., 2013). Previously, we have performed RNA-seq after RNAi of two HSP90 homologs (NlHsp90 and NlGRP94) in female BPH, and totally 52 and 41 differentially expressed genes were detected, respectively (results have not been published yet). We screened all the up- and down-regulated genes after RNAi treatment and found RNAi pathway was not affected or involved. So our results indicated that Hsc70/Hsp90 chaperones machinery might not affect RNAi pathway in this species.
In addition, we added in the discussion in L
“Recently, although it was demonstrated that Hsc70/Hsp90 chaperones machinery par-ticipate in RNAi pathways in some studies [27], RNAi has been used as a sufficient tool in many species to explore the in vivo function of HSPs. Our RNA-seq after RNAi of two HSP90 homologs (NlHsp90 and NlGRP94) in female BPH didn’t show interaction between HSP90 and RNAi pathways (unpublished data).”
Thank you for your providing the reference (Evgen’ev, M. B., Garbuz, D. G., & Zatsepina, O. G. 2014. Heat Shock Proteins and Whole Body Adaptation to Extreme Environments. doi:10.1007/978-94-017-9235-6). It has been cited as [27] in the Discussion.
References:
Xu H J , Chen T , Ma X F , et al. Genome-wide screening for components of small interfering RNA (siRNA) and micro-RNA (miRNA) pathways in the brown planthopper, Nilaparvata lugens (Hemiptera: Delphacidae).[J]. Insect Molecular Biology, 2013, 22(6):635-647.
Comment: 2) Parts 2.5 and 2.6: what are sample sizes for the treatments?
Response: Thanks. We have added the description of sample sizes for the treatments in Parts 2.5 and 2.6.
In L147-148, we added “One hundred 3rd instar nymphs were treated for each target genes, and each treatment was carried out for three biological replicates.”
In L157, we added “Each treatment of females was carried out for 10 biological replicates”
In L162-163, we added “In the fecundity assay, five treated females of each gene after mating with wild-type males were allowed to oviposit for 2 h in the same seedlings, then the rice seedlings…”
Reviewer 2 Report
In this study, the authors characterize the various paralogs of Hsp70 and DNAJ co-chaperones in the brown planthopper Nilaparvata lugens. They identified 9 distinct Hsp70 genes and 31 DNAJ genes and perform detailed phylogenetic analysis on these. They follow up on this very nicely by examining tissue-specific expression of these paralogs as well as their inducibility under thermal stress. Finally they demonstrate functional uniqueness of these paralogs through detailed phenotypic analysis after systematic knockdown of these genes. Overall, this is a well-written manuscript with clearly-formatted data. This work substantially contributes to the idea that despite chaperone and co-chaperone paralog similarity (and apparent redundancy), paralogs clearly have distinct cellular roles.
Specific comments:
- Although figure 1 shows a nice general exon/intron comparison, it would be nice to see a full amino acid alignment, at least for the Hsp70 paralogs.
- Some of the figures are so small that they become hard to read. Examples are figs. 1, 4, 7. Please expand if possible, although this may be fixed at the production stage by the journal.
- Some figures have mixtures of fonts. Please use arial throughout.
- Please incorporate the following refs for other Hsp70 diversity (PUBMED ID: 32284329, 32880065 and 31233900).
- Although the authors consider differences in function arising from expression and sequence differences, they do not consider differential post-translational modification of Hsp70. Please reference PUBMED ID: 32518165 in the discussion.
Author Response
Reviewer2
In this study, the authors characterize the various paralogs of Hsp70 and DNAJ co-chaperones in the brown planthopper Nilaparvata lugens. They identified 9 distinct Hsp70 genes and 31 DNAJ genes and perform detailed phylogenetic analysis on these. They follow up on this very nicely by examining tissue-specific expression of these paralogs as well as their inducibility under thermal stress. Finally they demonstrate functional uniqueness of these paralogs through detailed phenotypic analysis after systematic knockdown of these genes. Overall, this is a well-written manuscript with clearly-formatted data. This work substantially contributes to the idea that despite chaperone and co-chaperone paralog similarity (and apparent redundancy), paralogs clearly have distinct cellular roles.
Response: Thank you for your positive comments.
Comment: 1) Although figure 1 shows a nice general exon/intron comparison, it would be nice to see a full amino acid alignment, at least for the Hsp70 paralogs.
Response: Thank you for your advice. We added a full amino acid alignment of Hsp70 paralogs of BPH in 2.3 and Figure 1.
In L113-114, we added: “Amino acid sequence alignments were carried out with ClustalX and GeneDoc.”
In L87-90, we added: “A full amino acid alignment of NlHSP70 proteins in Figure 1C showed that NlHSP70 family members were highly conserved with three conserved regions (GIDLGTTYS, IFDLGGGTFDVSIL, and VGGSTRIPKVQ) defining HSP70 family signatures.”
The changed figure is shown as follows.
And we revised the legend of Figure 1 in L94-98:
Exon-intron organization of NlHSP70 (A) and NlDANJ (B) genes. The yellow box, blue box and gray line represent the CDS, untranslated region and intron, respectively. (C) A full amino acid sequence alignment of NlHSP70 proteins. Identical amino acid residues and conservative substitutions are shaded in black and gray, respectively. HSP70 family signatures are indicated with red boxes.
Comment: 2) Some of the figures are so small that they become hard to read. Examples are figs. 1, 4, 7. Please expand if possible, although this may be fixed at the production stage by the journal.
Response: Thank you. We have expanded the size of all the figures, and we also supply the original figures in the system.
Comment: 3) Some figures have mixtures of fonts. Please use arial throughout.
Response: Thanks. We have changed the fonts of all the figures with arial.
Comment: 4) Please incorporate the following refs for other Hsp70 diversity (PUBMED ID: 32284329, 32880065 and 31233900).
Response: Thanks. We incorporated the suggested references for other Hsp70 diversity.
In L508-512, we added: “Generally, it has been assumed that HSP70 paralogs have similar activities and are largely functionally redundant. However, accumulated studies have revealed the diversity of HSP70s. It has been demonstrated that different HSP70 isoforms have unique co-chaperone and client interactors, and functional diversity is defined by the affinities for specific co-chaperones.”
Comment: 5) Although the authors consider differences in function arising from expression and sequence differences, they do not consider differential post-translational modification of Hsp70. Please reference PUBMED ID: 32518165 in the discussion.
Response: Thank you for your advice. We added this part in the discussion in L588-593 as: “More recently, accumulated evidence has uncovered a vast array of post-translational modifications on Hsp70 family proteins, including phosphorylation, methylation, acetylation, ubiquitination, AMPylation, and ADP-ribosylation. As a result, the differences in function of HSP70/DNAJ machinery might arise from expression and sequence differences as well as differential post-translational modification of HSP70 isoforms.”
- Other revisions
In addition, we corrected some clerical errors in the manuscript.
1) L358: “Figure 7” was revised as “Figure 8”.
2) L405: “Figure 7” was revised as “Figure 9”.
3) L413: “3.5.4. Essential roles of NlDNAJs in BPH development” was revised as “3.5.4. Essential roles of NlDNAJs in female fecundity”.
Round 2
Reviewer 1 Report
I review the manuscript "HSP70/DNAJ family of genes in the brown planthopper, Nilaparvata lugens: diversity and function" for a seconds time. After the first round of review, I had a major concern about the experimental design being inappropriate as the genes chosen by the authors might be involved in the RNAi machinery itself, undermining the finding of the study. However, the authors confirmed that they have accomplished other experiments which allow discarding this possibility. The data is not published but I am happy with the way this point addressed in Discussion. The other minor corrections have been addressed as well.
I find the manuscript interesting, well-written and I am sure it will find its readers among researchers studying invasive biology, as well as interested in wider evolutionary questions. I recommend the manuscript for publishing in its present form.